# Calprotectin, Azurocidin, and Interleukin-8: Neutrophil Signatures with Diagnostic and Prognostic Value in Sepsis

**DOI:** 10.3390/biomedicines13112673

**Published:** 2025-10-30

**Authors:** Simona Gigliotti, Michele Manno, Francesca Divenuto, Grazia Pavia, Cinzia Peronace, Francesca Trimboli, Concetta Zangari, Valentina Tancrè, Francesca Greco, Manuela Colosimo, Pasquale Minchella, Luigi Principe, Nadia Marascio, Francesca Licata, Aida Bianco, Alessandro Russo, Federico Longhini, Angela Quirino, Giovanni Matera

**Affiliations:** 1Unit of Clinical Microbiology, Department of Health Sciences, “Magna Græcia” University of Catanzaro, “R. Dulbecco” Teaching Hospital, 88100 Catanzaro, Italy; s.gigliotti@unicz.it (S.G.); michele.manno@unicz.it (M.M.); francescadivenuto@unicz.it (F.D.); graziapavia@unicz.it (G.P.); cinzia.peronace@aourenatodulbecco.it (C.P.); trimboli@unicz.it (F.T.); concettazangari80@gmail.com (C.Z.); valentina.tancre@virgilio.it (V.T.); nmarascio@unicz.it (N.M.); mmatera@unicz.it (G.M.); 2Microbiology and Virology Unit, Annunziata Hospital, 87100 Cosenza, Italy; francesca.greco@aocs.it; 3Microbiology and Virology Unit, “R. Dulbecco” Teaching Hospital, 88100 Catanzaro, Italy; manuela.colosimo@aourenatodulbecco.it (M.C.); pasquale.minchella@aourenatodulbecco.it (P.M.); 4Clinical Microbiology and Virology Unit, Great Metropolitan Hospital “Bianchi-Melacrino-Morelli”, 89133 Reggio Calabria, Italy; luigi.principe@ospedalerc.it; 5Department of Health Sciences, School of Medicine, “Magna Græcia” University Hospital, 88100 Catanzaro, Italy; f.licata@unicz.it; 6Department of Medical and Surgical Sciences, “Magna Græcia” University of Catanzaro, “R. Dulbecco” Teaching Hospital, 88100 Catanzaro, Italy; a.bianco@unicz.it; 7Infectious and Tropical Disease Unit, Department of Medical and Surgical Sciences, “Magna Græcia” University of Catanzaro, 88100 Catanzaro, Italy; a.russo@unicz.it; 8Anesthesia and Intensive Care Unit, Department of Medical and Surgical Sciences, “Magna Græcia” University of Catanzaro, “R. Dulbecco” Teaching Hospital, 88100 Catanzaro, Italy; flonghini@unicz.it

**Keywords:** inflammatory biomarkers, septic patients, cytokines

## Abstract

**Background:** Sepsis remains a major cause of morbidity and mortality in both developed and limited-resource countries. Despite over a century of research, accurate biomarkers for reliable diagnosis and prognosis in critically ill patients have yet to be established. **Methods:** This multicenter retrospective observational study aims to evaluate serum levels of Calprotectin, Azurocidin, cytokines, chemokines, procalcitonin (PCT) and C-Reactive Protein (CRP) in 15 healthy volunteers (controls), 15 non-infectious SIRS patients, 92 alive septic patients (Sepsis_A) and 29 dead septic patients (Sepsis_D). **Results:** Most biomarkers showed significantly higher serum concentrations in septic patients compared with controls, with IL-4 being increased only in the Sepsis_D group. In addition, several markers, including Calprotectin, Azurocidin, IL-6, IL-8, IL-10, TNF-α, and IL-35, were progressively elevated from SIRS to Sepsis_A and Sepsis_D cohorts, reflecting disease severity. All biomarkers showed good diagnostic performance for predicting Gram-negative bacteremia, although their accuracy in discriminating survivors from non-survivors was relatively low. **Conclusions:** In conclusion, calprotectin, azurocidin, IL-8, TNF-α, and IL-35 may assist clinicians in identifying Gram-negative bacteremia in septic patients; however, their prognostic value appears to be limited.

## 1. Introduction

Sepsis is one of the leading causes of death worldwide. It is still difficult to diagnose the infection quickly and accurately, and the diagnosis of sepsis is often not timely. Although more than 250 biomarkers have been studied over the last years, no biomarkers accurately differentiate between sepsis and sepsis-like syndrome. It is necessary to find one or more biomarkers useful to assist clinicians in the diagnosis and prognosis of sepsis. Herein, we provide current data on the clinical utility of host-response biomarkers, offer guidance on optimizing their use, and propose the need for future research [1].

The lack of a biomarker that serves as a generally accepted gold standard for sepsis is a common limitation of all sepsis studies. Biomarkers can play a key role in the timely diagnosis and management of sepsis. C-reactive protein (CRP) and procalcitonin (PCT) are widely used biomarkers, but their diagnostic accuracy has been questioned [2]. Indeed, they cannot easily distinguish infection from inflammation. Therefore, biomarkers with high diagnostic sensitivity and specificity are indispensable [3]. Azurocidin appears to play an important role in the pathophysiology of severe bacterial infections, thus representing a potential diagnostic marker and target for the treatment of sepsis [4]. Calprotectin is an acute-phase protein released into the circulation by monocytes and neutrophils that is thought to be more sensitive than CRP in detecting minimal residual inflammation [5]. Such polypeptide can stimulate the TLR4 receptor on several types of host cells, and such a feature would initiate an inflammatory cascade, relevant to sepsis pathogenesis [6,7].

Following the pioneering work of Bone et al. [8], which primarily identified pathogenic mediators and sepsis biomarkers of macrophage/monocyte origin, Hotchkiss et al. later shifted the focus to alterations in lymphocyte counts and to the diverse mediators derived from distinct lymphocyte subsets (Th1, Th2, Treg, and Breg) [9].

Neutrophils have been reported to play a major role in the pathogenesis of sepsis, both through the direct release of mediators and by modulating the activity of other immune cells [10]. Therefore, we evaluated mediators released from neutrophil granules (Calprotectin and azurocidin) and neutrophil chemokine (IL-8) to address the critical role of neutrophil mediators as potential sepsis biomarkers. Moreover, several groups of cytokines (Th1, Th2, Treg, Breg) were also evaluated as potential sepsis biomarkers. Also, IL-35, a novel Breg cytokine, has been rarely assessed as a sepsis biomarker. Those few publications, which include IL-35, never focused specifically on the diagnostic or prognostic role of such cytokine in sepsis and on the possibility of being used as a helpful tool to assist clinicians’ decisions [11,12].

We therefore designed this multicenter, retrospective observational study to assess serum levels of calprotectin, azurocidin, PCT, CRP, Interleukins and Tumor Necrosis Factor α (TNF-α), in healthy controls, patients with non-infectious Systemic Inflammatory Response Syndrome (SIRS), or sepsis/septic shock with blood cultures positive for Gram-negative bacteria. In addition, we aimed to determine whether these biomarkers possess diagnostic value (for identifying the presence or absence of sepsis) and prognostic value (for predicting survival or not).

## 2. Materials and Methods

The Ethics Committee of the Calabria Region approved the study protocol (approval number 128/2023, on 22 December 2023); given the retrospective study design, written informed consent was waived. All procedures were conducted in accordance with the principles of the Declaration of Helsinki. The anonymized dataset generated and analyzed during the current study is available from the corresponding author upon reasonable request.

This multicenter, retrospective observational study was conducted from February to June 2024. We enrolled only adult participants (i.e., >18 years old), classified into the following groups: (1) 15 healthy volunteers; (2) 15 patients with non-infectious SIRS; (3) 92 alive septic patients (Sepsis_A), as defined by the Third International Consensus Definitions for Sepsis and Septic Shock (Sepsis-3) [13]; and (4) 29 non-survivor (i.e., died) septic patients (Sepsis_D). Both Sepsis_A and Sepsis_D patients were required to have blood cultures positive for Gram-negative bacteria. Exclusion criteria included malignancies, alcohol or drug abuse, pregnancy, and lactating women.

Non-infectious SIRS, defined as an exaggerated defense response of the body to a noxious stressor (including trauma, surgery, acute inflammation or ischemia/reperfusion), was diagnosed by the presence of two or more of the following clinical criteria in the absence of infection: body temperature > 38 °C or <36 °C; heart rate > 90 beats/min; respiratory rate > 20 breaths/min or arterial PaCO_2_ < 32 mmHg; and white blood cell count > 12 × 10^9^/L, < 4 × 10^9^/L, or > 10% immature (band) forms [14].

For all participants, blood samples for biomarker assessment were obtained at hospital admission, corresponding to the time of SIRS or sepsis diagnosis, before the initiation of antibiotic treatment. Mortality was determined at 28 days following SIRS diagnosis or, in the case of sepsis, 28 days after the first positive blood culture result.

### 2.1. Serum Biomarker Assessment

Serum Calprotectin levels were measured using the Calprest NG-S (ECL) ELISA kit (Eurospital Diagnostic, Trieste, Italy). This enzyme immunoassay employs colorimetric detection based on polyclonal and monoclonal antibodies directed against Calprotectin. The immobilized antibody captures Calprotectin from the diluted serum sample within the wells. Subsequently, peroxidase-conjugated (HRP) antibodies bind to the captured Calprotectin, and the enzyme catalyzes the conversion of the substrate into a colored product. The color intensity is proportional to the amount of conjugate bound, and therefore to the Calprotectin concentration. Serum Calprotectin levels were calculated by interpolation from a calibration curve. After centrifugation of the primary collection tube (3000× *g* for 10 min at room temperature), serum samples were obtained within 4–6 h of collection. A minimum volume of 500 µL was required for analysis. Samples not tested immediately were stored at −20 °C in 0.5–2 mL screw-capped tubes and were not subjected to more than three freeze–thaw cycles. The assay measuring range was 0–150 ng/mL. Intra- and inter-assay coefficients of variation on serum samples were <7% and <12.2%, as per the manufacturer’s documentation.

Standard and serum samples were added to microplate wells coated with a biotin-conjugated antibody specific for Azurocidin (AZU1) (Enzyme-linked Immunosorbent Assay Kit for AZU1, Cloud-Clone Corp., Katy, TX, USA). Avidin conjugated to horseradish peroxidase (HRP) was then added, and the plate was incubated. After the addition of the TMB substrate solution, only wells containing AZU1 bound to the biotin-conjugated antibody and enzyme-conjugated Avidin developed a color change. The enzyme–substrate reaction was stopped by adding sulfuric acid solution, and the resulting color intensity was measured spectrophotometrically at 450 ± 10 nm. The concentration of AZU1 in each sample was determined by comparing the optical density (O.D.) values to the standard calibration curve. The assay measuring range was 0.312–20 ng/mL. Intra- and inter-assay coefficients of variation were <10% and <12%, as per the manufacturer’s documentation.

Clinical biochemistry data regarding soluble serum levels factors were detected by chemiluminescence assay (CLIA) for PCT and CRP [15].

To quantify serum concentrations of IL-4, IL-6, IL-8, IL-10, and TNF-α, samples were analyzed using a biochip array incorporating specific primary antibodies (Cytokine and Growth Factors Array, Randox Laboratories, Crumlin, UK), following the manufacturer’s instructions. Briefly, serum samples were incubated on the biochip for 1 h at 37 °C, followed by washing and a subsequent 1 h incubation with a horseradish peroxidase (HRP)-conjugated secondary antibody. Chemiluminescent signals were then detected using the Evidence Investigator biochip analyzer (Randox Laboratories, Crumlin, UK) with the Cytokines Array I and High Sensitivity kit and quantified with the dedicated Randox software [16]. The assay measuring range was 0–450 pg/mL for IL-4, 0–400 pg/mL for IL-6, 0–1450 pg/mL for IL-8, 0–450 pg/mL for IL-10, and 0–600 pg/mL for tumor necrosis factor (TNF)-α. Intra- and inter-assay coefficients of variation were9.5% and 11.8% for IL-4,11.9% and 8.4% for IL-6, 9.4% and 9.2% for IL-8, 5.6% and 6.5% for IL-10, and 7.1% and 6.7% for TNF-α, respectively [16].

Human IL-35 levels were measured using a sandwich ELISA kit (Human IL-35 ELISA Kit, Assay Genie, Dublin, Ireland). The ELISA plate was pre-coated with an antibody specific for human IL-35. Standards and serum samples were added to the wells and allowed to bind to the immobilized antibodies. Subsequently, a biotinylated detection antibody specific for human IL-35 and an avidin–horseradish peroxidase (HRP) conjugate were added sequentially and incubated. After washing to remove unbound components, a substrate solution was added; wells containing the biotinylated detection antibody and avidin–HRP conjugate developed a blue color. The reaction was terminated by adding stop solution, resulting in a color change from blue to yellow. Optical density (OD) was measured spectrophotometrically at 450 nm, and IL-35 concentrations were determined by interpolating the sample OD values from the standard calibration curve. The detection range of the assay was 15.63–1000 pg/mL. Intra- and inter-assay coefficients of variation were <8% and <10%, as per manufacturer’s documentation.

All laboratory personnel performing the biomarker analyses were blinded to the patients’ clinical status and outcomes. Samples were processed and analyzed according to standardized procedures, and no clinical information was available to the operators at any stage of the laboratory workflow. This approach was adopted to minimize bias and ensure the objectivity of the measurements.

### 2.2. Statistical Analysis

Continuous variables are expressed as mean ± standard deviation (SD) or as median [25th–75th percentile], while categorical variables are presented as percentages. The normality of data distribution was assessed using the Kolmogorov–Smirnov test. Given the non-parametric distribution of the data, the Kruskal–Wallis test was used to assess differences between groups. Multiple pairwise comparisons were performed using the corrected Dunn’s test.

Sensitivity, specificity, positive (PLR) and negative (NLR) likelihood ratios, and positive (PPV) and negative (NPV) predictive values were computed for all biomarkers for both diagnostic and prognostic evaluations. Receiver Operating Characteristic (ROC) curves and the corresponding areas under the curve (AUCs) with 95% confidence intervals (95% CI) were calculated using the nonparametric approach described by DeLong et al. [17]. The Youden index (J = max [sensitivity + specificity − 1]) was applied to determine the optimal cut-off values. *p*-values of less than 0.05 were considered statistically significant. Statistical analysis was done using STATA software program, version 18.

## 3. Results

The demographic characteristics of the study population are summarized in Table 1. The cohort comprised 15 healthy volunteers, 15 patients with non-infectious SIRS, and 92 and 29 patients assigned to the Sepsis_A and Sepsis_D groups, respectively. Gender and age were comparable among groups, except for the healthy control group, which consisted of younger individuals and a higher proportion of females (80%) (Table 1). Some baseline imbalances were observed between the SIRS and Sepsis groups, particularly regarding the admission diagnosis and the need for organ support, including mechanical ventilation, vasoactive agents, renal replacement therapy, and/or extracorporeal membrane oxygenation (ECMO). The source of infection and the isolated pathogens were similar between the Sepsis_A and Sepsis_D cohorts; however, SOFA and APACHE II scores were higher in the latter group compared to the former, as expected.

### 3.1. Serum Biomarkers

Table 2 reports the median [IQR] values of the measured serum biomarkers together with the corrected *p*-values obtained from multiple group comparisons, whereas Figure 1 displays the corresponding boxplots with exact *p*-values. All biomarkers, except IL-4, were significantly increased in the Sepsis_A and Sepsis_D cohorts compared with Controls. Specifically, IL-4 levels were higher only in Sepsis_D patients (*p* = 0.0153) but not in Sepsis_A patients (*p* = 0.2590).

CRP (*p* = 0.0003) and IL-6 (*p* = 0.0176) were also significantly elevated in the SIRS cohort compared with Controls. When comparing SIRS and Sepsis_A patients, higher concentrations were observed in the latter group for Azurocidin (*p* = 0.0137), PCT (*p* = 0.0114), IL-8 (*p* = 0.0153), TNF-α (*p* = 0.0192), and IL-35 (*p* < 0.0001).

Compared with the SIRS cohort, Sepsis_D patients exhibited higher levels of Calprotectin (*p* = 0.0072), Azurocidin (*p* < 0.0001), IL-4 (*p* = 0.0112), IL-8 (*p* = 0.0015), IL-10 (*p* = 0.0007), TNF-α (*p* = 0.0198), and IL-35 (*p* < 0.0001). Finally, the Sepsis_D cohort showed significantly higher concentrations of Azurocidin (*p* = 0.0009) and IL-6 (*p* = 0.0452) compared with Sepsis_A patients.

### 3.2. Diagnostic and Prognostic Values

The predictive performance of all biomarkers for Gram-negative bacteremia was evaluated, and the results of the ROC analyses are reported in Table 3, with the corresponding curves shown in Figure 2. All biomarkers demonstrated good diagnostic accuracy for Gram-negative bacteremia, with higher AUC values observed for Calprotectin (0.864 [0.791–0.919]), Azurocidin (0.808 [0.739–0.867]), PCT (0.908 [0.842–0.953]), IL-8 (0.919 [0.842–0.966]), IL-10 (0.812 [0.716–0.887]), TNF-α (0.903 [0.822–0.955]), and IL-35 (0.910 [0.848–0.953]) (*p* < 0.001 for all ROC curves).

The prognostic performance of the same biomarkers for mortality prediction was also assessed, with results presented in Table 3 and the corresponding ROC curves illustrated in Figure 3. All biomarkers except PCT were able to predict mortality, although the AUC values were lower than those observed in the diagnostic analyses.

## 4. Discussion

In our study, serum concentrations of calprotectin, azurocidin, PCT, CRP, interleukins, and TNF-α were generally increased in septic patients, regardless of outcome, compared with healthy controls. However, their patterns in SIRS patients and between the Sepsis_A and Sepsis_D groups varied for each biomarker analyzed. Moreover, all biomarkers demonstrated good diagnostic performance for Gram-negative bloodstream infection, whereas their ability to predict mortality was relatively limited.

In this study, we sought to investigate distinct classes of sepsis biomarkers, stratified according to their functional characteristics. We first focused on Th1-associated mediators, including TNF-α and IL-6. Th1 markers are typically associated with the proinflammatory response and have been reported to correlate with a more severe course of sepsis and poorer clinical outcomes [18].

TNF-α levels among septic patients were significantly higher compared with both controls and SIRS cohorts, supporting the diagnostic value of this cytokine, as also confirmed by the ROC analysis. Conversely, TNF-α concentrations did not differ significantly between the Sepsis_A and Sepsis_D cohorts, suggesting a limited prognostic role, consistent with our ROC findings and previous reports in the literature [19].

Based on our results, IL-6 was able to discriminate both SIRS and septic patients from controls, as well as to differentiate between the Sepsis_A and Sepsis_D cohorts. These findings suggest that IL-6 may serve as both a diagnostic and prognostic biomarker in sepsis [20]. The ROC analysis further confirmed the strong predictive value of IL-6 for the diagnosis of Gram-negative bloodstream infection and for outcome prediction. Therefore, IL-6 appears to represent a promising diagnostic and prognostic tool that could assist clinicians in clinical decision-making [21,22].

Our data showed that IL-4 levels were significantly higher in Sepsis_D patients compared with healthy controls and SIRS, whereas differences among controls, SIRS, and Sepsis_A patients were not statistically significant. The absence of significant differences between SIRS and Sepsis_A groups suggests that IL-4 alone may have limited diagnostic value in distinguishing sepsis from non-infectious SIRS. Recent literature has highlighted the often-contradictory role of IL-4 in sepsis [23]. Moreover, IL-4 has been associated with immunoparalysis and partial stimulation of certain proinflammatory cytokines, such as IL-6 and TNF-α, suggesting a potentially modulatory and even protective role in the septic response [23]. The diagnostic and prognostic roles of IL-4 in human sepsis have been poorly investigated to date. However, available evidence suggests that IL-4 is more closely associated with Gram-positive bacteremia than with sepsis caused by Gram-negative bacteria or fungi. As a Th2 cytokine, IL-4 may contribute diagnostically to distinguishing between SIRS and bacteremic patients. Nevertheless, our data indicate that IL-4 is not a reliable biomarker for assessing disease progression or predicting clinical outcomes in septic patients.

Among inhibitory cytokines, IL-10 was evaluated in our study, and the most relevant finding was the significantly higher levels observed in Sepsis_D patients compared with Sepsis_A. This suggests that IL-10 may serve as a potential biomarker for estimating mortality risk among septic patients [24]. Conversely, the absence of significant differences in IL-10 levels between Sepsis_A and SIRS groups indicates that this cytokine may have limited diagnostic value in distinguishing early sepsis from non-infectious SIRS. This finding could be explained by the predominant role of IL-10 during the later stages of sepsis, whereas our study evaluated patients primarily during the early phase of the disease.

Based on the ROC analysis for both diagnostic and prognostic evaluations, IL-10 appears to be a useful biomarker for predicting Gram-negative bloodstream infection, as previously reported [25], and for estimating mortality risk.

Calprotectin has been reported as a pivotal mediator during the immunosuppressive stage of late sepsis [26], an effect that may also be mediated by IL-10 [27]. A novel finding of our study is the potential involvement of both calprotectin and IL-10 in this late immunosuppressive phase of sepsis. However, direct evidence for this mechanism remains limited, and further dedicated studies are warranted to specifically investigate this aspect.

From a mechanistic perspective, the dual and seemingly opposite roles of calprotectin may result from the involvement of different innate immune cells active during distinct stages of sepsis. In the acute phase, calprotectin is predominantly associated with the activity of dendritic cells and neutrophils, whereas in the immunosuppressive phase, its effects are mainly mediated by myeloid-derived suppressor cells (MDSCs), which play a key role in sustaining immune dysfunction during late sepsis [26,28,29]. In addition, calprotectin has been reported to act as a damage-associated molecular pattern (DAMP) during sepsis [30], exerting its effects through the Toll-like receptor 4 (TLR4) signaling complex [31].

Data on IL-35 showed that its serum concentrations were significantly higher in both groups of septic patients compared with healthy controls and SIRS subjects. However, no significant differences were observed between the Sepsis_A and Sepsis_D cohorts. Consistent with previous reports [11], IL-35 demonstrated good predictive value for the diagnosis of Gram-negative bacteremia, whereas its ability to predict mortality was limited. This pattern, in which IL-35 serves as a prominent diagnostic marker but lacks prognostic utility, suggests its involvement in the early proinflammatory phase of sepsis. Moreover, IL-35 may act during the initial immune response, potentially in coordination with neutrophil activation at the site of infection, as suggested by recent studies linking IL-35 release to neutrophil activity [32].

Regarding IL-8, a noteworthy finding was the significant difference in IL-8 concentrations between septic and non-septic cohorts. Although the role of IL-8 in sepsis has been investigated for more than three decades [33], conclusive and consistent evidence remains limited [34]. Holub et al. reported elevated IL-8 levels in septic patients, and more recently, increasing evidence has highlighted both the pathogenetic and diagnostic roles of neutrophil-derived mediators, for which IL-8 serves as a key chemotactic factor. Consistent with these observations, our ROC analysis demonstrated that IL-8 was a reliable predictor for the diagnosis of Gram-negative bloodstream infection, whereas it did not show prognostic significance.

We also found that azurocidin levels were significantly higher in septic patients compared with both SIRS patients and healthy controls. Furthermore, the Sepsis_D cohort exhibited significantly higher azurocidin concentrations than the Sepsis_A cohort. ROC analysis demonstrated good diagnostic and prognostic performance of azurocidin in our population, consistent with previous reports [35,36].

We also investigated two reference biomarkers, PCT and CRP, used for many years to assist clinicians in the diagnosis of sepsis. Regarding PCT, our data suggest a diagnostic role of PCT in septic patients. On the other hand, CRP has a minor role in both diagnosis and prognosis of this population, as previously reported by a similar study [3].

Finally, it should be noted that calprotectin, azurocidin, IL-4, and IL-8, although acting through different mechanisms, may all contribute significantly to the innate defense against invading pathogens. The binding of azurocidin to pathogen-associated molecules such as lipopolysaccharide (LPS) may facilitate pathogen clearance by professional phagocytes. Conversely, calprotectin exerts antimicrobial activity through the chelation of essential nutrient metals (Zn^2+^, Cu^2+^), a process known as nutritional immunity [37,38].

Of note, the elevation of several biomarkers observed in our study could also be influenced by patients’ comorbidities and by the overall severity of illness. Chronic conditions such as diabetes, cardiovascular disease, or renal dysfunction are known to sustain a proinflammatory state, which may contribute to higher baseline levels of cytokines and acute-phase proteins [39,40,41]. Likewise, disease severity, reflected by higher SOFA and APACHE II scores, is typically associated with a more pronounced systemic inflammatory response and organ dysfunction [14,42], which could amplify biomarker concentrations independently of infection status. Therefore, these factors should be considered when interpreting biomarker elevations in septic patients.

### Strengths and Limitations

Before drawing our conclusions, several strengths and limitations of this study should be discussed.

One notable strength is its multicenter design, which enhances the robustness of the data and supports the generalizability of our findings to a broader population of patients [43,44,45]. Furthermore, few studies to date have simultaneously investigated novel diagnostic and prognostic biomarkers alongside established and widely recognized sepsis biomarkers within the same patient cohort. This comparative approach provides valuable insight into the relative clinical performance of emerging biomarkers and strengthens the translational relevance of our results.

However, this study also has several limitations that should be acknowledged. First, its retrospective observational design may be affected by selection bias, and the findings may not be fully representative of the general population. In addition, the presence of potential confounding variables limits the ability to establish causal relationships [46]. Second, biomarker measurements were obtained at a single time point, which prevented the assessment of dynamic changes during the course of illness that might provide additional prognostic information. Furthermore, as this was a retrospective study, the exact timing of sample collection in relation to symptom onset, hospital admission, and blood culture positivity could not be systematically retrieved for all patients. Nevertheless, all samples were obtained at the time of hospital admission and before the initiation of antibiotic therapy, as verified in the clinical records. This standardized sampling procedure helps to limit variability associated with timing and ensures consistency across the study cohort. We acknowledge, however, that the absence of precise temporal data may limit the detailed interpretation of biomarker dynamics. Future prospective investigations with predefined and recorded time points are warranted to better assess the impact of sampling timing on biomarker performance and reproducibility. To address these limitations, we have already planned a further study aimed at evaluating biomarker kinetics in this patient population. Third, this study exclusively enrolled septic patients with an initial Gram-negative bloodstream infection, as detailed in the Materials and Methods section. Consequently, the present findings primarily characterize the host response and biomarker dynamics in Gram-negative sepsis and may not be fully generalizable to sepsis of Gram-positive or fungal etiology. This limitation should be taken into account when interpreting the results. Finally, although the overall sample size was adequate for the primary analyses [3,47,48], the number of patients in some subgroups was limited, potentially reducing the statistical power to detect more subtle associations. These limitations should be taken into account when interpreting our findings and highlight the need for future prospective, longitudinal studies, for further validation, with larger and more balanced cohorts to confirm and expand upon our results.

## Figures and Tables

**Figure 1 biomedicines-13-02673-f001:**
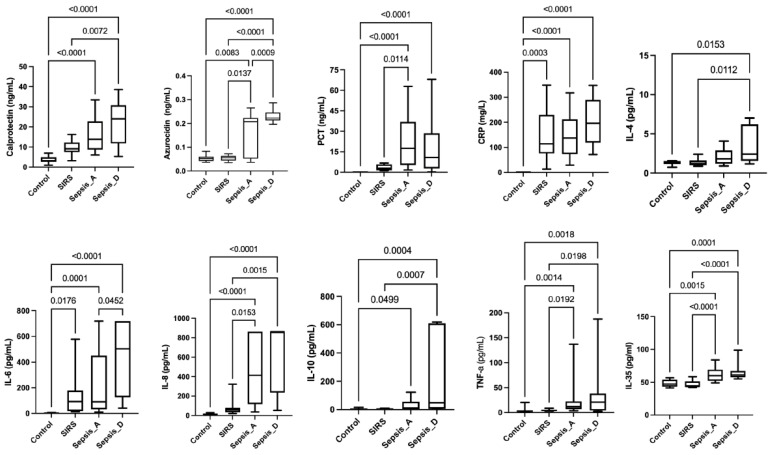
The figure depicts boxplots and exacted *p*-values for multiple comparisons between cohorts of patients in all the analyzed biomarkers. The bottom and top of the box indicate the 25th and 75th percentiles, the horizontal band near the middle of the box is the median, and the ends of the whiskers represent the 10th and 90th percentiles.

**Figure 2 biomedicines-13-02673-f002:**
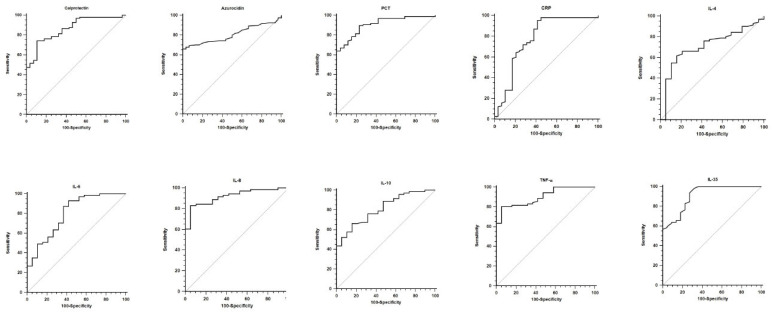
ROC analyses for diagnostic accuracy for Gram-negative bacteremia are depicted for every biomarker. See text and Table 3 for further explanation and data.

**Figure 3 biomedicines-13-02673-f003:**
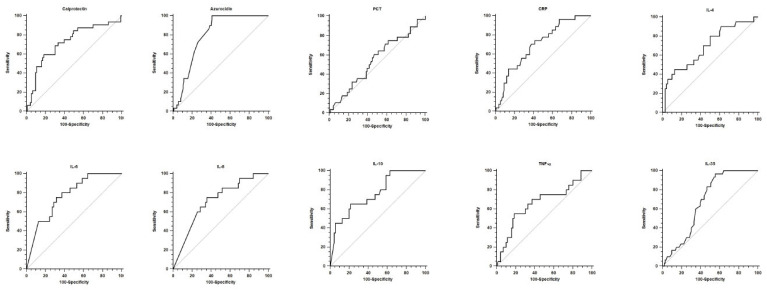
ROC analyses for prognostic accuracy for mortality are depicted for every biomarker. See text and Table 3 for further explanation and data.

**Table 1 biomedicines-13-02673-t001:** Characteristics of the cohorts included in the study.

	*Control (n = 15)*	*SIRS (n = 15)*	*Sepsis_A (n = 92)*	*Sepsis_D (n = 29)*
Age-(yr)	37.5 ± 15.3	59.5 ± 14.0	68.7 ± 15.5	71.2 ± 13.7
Male sex–*n* (%)	3 (20)	8 (53)	52 (57)	14 (48)
SOFA score	0 [0; 0]	4 [3; 6]	7 [5; 9]	10 [8; 12]
APACHE-II	0 [0; 0]	15 [8; 18]	18 [12; 21]	23 [16; 30]

*Comorbidities-n* (%)				
Chronic Respiratory Failure		0 (0)	7 (7)	3 (10)
Cardiovascular disease		1 (7)	18 (19)	8 (28)
Arterial Hypertension		4 (27)	38 (41)	15 (52)
Diabetes		2 (13)	28 (30)	11 (38)
Others		0 (0)	8 (8)	4 (14)

*Admission department-n* (%)				
Intensive Care Unit		3 (20)	37 (40)	26 (90)
Surgical wards		8 (53)	21 (23)	1 (3)
Medical wards		4 (27)	34 (37)	2 (7)

*Pathology of admission-n* (%)				
Respiratory		0 (0)	47 (51)	16 (56)
Cardiovascular		0 (0)	16 (17)	7 (24)
Neurologic		0 (0)	7 (8)	2 (7)
Trauma		10 (67)	12 (13)	3 (10)
Others		5 (33)	10 (11)	1 (3)

*Organ support requirments-n* (%)				
Mechanical ventilation		2 (13)	36 (39)	26 (90)
Vasoactive drugs		2 (13)	35 (38)	25 (86)
Renal replacement therapy		1 (7)	16 (17)	20 (69)
ECMO		0 (0)	1 (1)	2 (7)

*Source of infection–n* (%)				
Lung			50 (55)	18 (63)
Abdomen			3 (3)	1 (3)
Blood			22 (24)	6 (21)
Endocarditis			14 (15)	3 (10)
Urinary tract			3 (3)	1 (3)

*Gram-negative pathogen isolations-n* (%)				
*Escherichia coli*			27 (30)	10 (34)
*Klebsiella pneumoniae*			34 (37)	15 (52)
*Acinetobacter baumannii*			14 (15)	3 (10)
*Pseudomonas aeruginosa*			14 (15)	7 (24)
Others bacilli			3 (3)	1 (3)

Data pertaining to the groups of individuals are reported as mean ± standard deviation, median [IQR], or number (percentage). Please note that in Sesis_D group, some patients had more than 1 Gram-negative pathogen isolation. SOFA, Sequential Organ Failure Assessment; APACHE, Acute Physiologic Assessment and Chronic Health Evaluation II; ECMO, ExtraCorporeal Membrane Oxygenation.

**Table 2 biomedicines-13-02673-t002:** Serum biomarkers.

	Control (*n* = 15)	SIRS (*n* = 15)	Sepsis_A (*n* = 92)	Sepsis_D (*n* = 29)	Kruskal–Wallis *p*-Values (H Statistic)	Dunn’s Test *p*-Values
** *Calprotectin* ** **(ng/mL)**	3.19 [2.60; 4.88]	9.12 [7.47; 12.28]	13.81 [8.69; 22.71]	24.04 [11.76; 30.83]	<0.001 (H = 43.70)	0.1125 <0.0001 <0.0001 0.2169 0.0072 0.2827
** *Azurocidin* ** ** (ng/mL) **	0.052 [0.042; 0.060]	0.056 [0.044; 0.064]	0.208 [0.052; 0.223]	0.223 [0.213; 0.246]	<0.001 (H = 41.63)	>0.9999 0.0083 <0.0001 0.0137 <0.0001 0.0009
** *PCT* ** ** (ng/mL) **	0.10 [0.10; 0.20]	2.92 [1.32; 5.76]	17.5 [5.35; 36.9]	10.8 [2.8; 28.6]	<0.001 (H = 46.43)	0.2291 <0.0001 <0.0001 0.0114 0.1998 >0.9999
** *CRP* ** ** (mg/L) **	1 [1; 2]	114 [76; 230]	138 [73; 212]	196 [119; 289]	<0.001 (H = 42.06)	0.0003 <0.0001 <0.0001 >0.9999 0.6679 0.1998
** *IL-4* ** ** (pg/mL) **	1.28 [1.17; 1.55]	1.22 [1.01; 1.62]	1.82 [1.20; 2.89]	2.43 [1.56; 6.22]	0.008 (H = 14.53)	>0.9999 0.2590 0.0153 0.2374 0.0112 0.3702
** *IL-6* ** ** (pg/mL) **	1.40 [1.05; 2.58]	92.7 [18.4; 178.7]	90.7 [31.5; 449.7]	502.1 [126.4; 719.0]	<0.001 (H = 32.06)	0.0176 0.0001 <0.0001 >0.9999 0.1258 0.0452
** *IL-8* ** ** (pg/mL) **	12.93 [9.10; 17.77]	57.3 [31.8; 78.7]	413.5 [116.2; 862.0]	862.0 [235.9; 862.0]	<0.001 (H = 36.69)	0.8317 <0.0001 <0.0001 0.0153 0.0015 0.9207
** *IL-10* ** ** (pg/mL) **	0.55 [0.38; 7.70]	2.25 [0.99; 4.69]	9.86 [2.22; 56.2]	48.4 [4.79; 609.9]	<0.001 (H = 23.50)	>0.9999 0.0499 0.0004 0.0937 0.0007 0.0813
** *TNF-a* ** ** (pg/mL) **	2.77 [2.43; 3.96]	4.22 [3.46; 4.68]	11.71 [6.92; 22.35]	23.10 [5.19; 52.7]	<0.001 (H = 22.33)	>0.9999 0.0014 0.0018 0.0192 0.0198 >0.9999
** *IL-35* ** ** (pg/mL) **	47.35 [43.30; 53.88]	43.67 [43.30; 51.40]	60.12 [52.13; 69.29]	61.34 [57.68; 67.14]	<0.001 (H = 39.22)	>0.9999 0.0015 0.0001 <0.0001 <0.0001 0.7527

Data are expressed as median [25th; 75th percentile]. *p*-values were derived using analysis of variance on ranks for non-repeated measures (Kruskal–Wallis test) and adjusted for multiple pairwise comparisons among groups using Dunn’s test. The second-to-last and last columns of the table report these corrected *p*-values, respectively. A: SIRS vs. Control; B: Sepsis_A vs. Control; C: Sepsis_D vs. Control; D: Sepsis_A vs. SIRS; E: Sepsis_D vs. SIRS; F: Sepsis_D vs. Sepsis_A. PCT, Procalcitonin; CRP, C-Reactive Protein; IL, Interleukin; TNF-a, Tumor Necrosis Factor a.

**Table 3 biomedicines-13-02673-t003:** Diagnostic and prognostic accuracy for all biomarkers.

	AUC	*p* Value	Cut-Off	Sensitivity	Specificity	PLR	NLR	PPV (%)	NPV (%)
** *Diagnostic accuracy* **									
*Calprotectin*	0.864 [0.791–0.919]	<0.001	10.04	74.2 [64.3–82.6]	89.3 [71.8–97.7]	6.93 [2.4–20.3]	0.29 [0.2–0.4]	96.0 [88.8–99.2]	50.0 [35.4–64.6]
*Azurocidin*	0.808 [0.739–0.867]	<0.001	0.102	66.1 [57.0–74.5]	100.0 [88.4–100.0]	//	0.34 [0.3–0.4]	100.0 [95.5–100.0]	42.3 [30.6–54.6]
*PCT*	0.908 [0.842–0.953]	<0.001	1.44	89.7 [81.9–94.9]	76.9 [56.4–91.0]	3.89 [1.9–7.9]	0.13 [0.07–0.3]	93.5 [86.5–97.6]	66.7 [47.2–82.7]
*CRP*	0.777 [0.698–0.844]	<0.001	20.4	95.3 [89.4–98.5]	58.6 [38.9–76.5]	2.30 [1.5–3.6]	0.08 [0.03–0.2]	89.5 [82.3–94.4]	77.3 [54.6–92.2]
*IL-4*	0.715 [0.610–0.805]	0.007	1.57	62.0 [49.7–73.2]	84.2 [60.4–96.6]	3.92 [1.4–11.3]	0.45 [0.3–0.6]	93.6 [82.3–98.7]	37.2 [23.0–53.3]
*IL-6*	0.799 [0.701–0.876]	<0.001	13.79	93.0 [84.3–97.7]	57.9 [33.5–79.7]	2.21 [1.3–3.8]	0.12 [0.05–0.3]	89.2 [79.8–95.2]	68.7 [41.3–89.0]
*IL-8*	0.919 [0.842–0.966]	<0.001	92.37	83.1 [72.3–91.0]	94.7 [74.0–99.9]	15.79 [2.3–106.7]	0.18 [0.1–0.3]	98.3 [91.1–100.0]	60.0 [40.6–77.3]
*IL-10*	0.812 [0.716–0.887]	<0.001	4.87	66.2 [54.0–77.0]	84.2 [60.4–96.6]	4.19 [1.5–12.0]	0.40 [0.3–0.6]	94.0 [83.5–98.7]	40.0 [24.9–56.7]
*TNF-a*	0.903 [0.822–0.955]	<0.001	5.41	80.3 [69.1–88.8]	94.7 [74.0–99.9]	15.25 [2.3–103.1]	0.21 [0.1–0.3]	98.3 [90.8–100.0]	56.2 [37.7–73.6]
*IL-35*	0.910 [0.848–0.953]	<0.001	48.83	93.7 [87.4–97.4]	72.7 [49.8–89.3]	3.44 [1.7–6.8]	0.09 [0.04–0.2]	94.5 [88.5–98.0]	69.6 [47.1–86.8]

** *Prognostic accuracy* **									
*Calprotectin*	0.721 [0.634–0.798]	<0.001	20.89	59.4 [40.6–76.3]	81.7 [72.4–89.0]	3.25 [1.9–5.4]	0.50 [0.3–0.8]	52.8 [35.2–69.8]	85.4 [76.3–92.0]
*Azurocidin*	0.796 [0.723–0.857]	<0.001	0.18	100.0 [88.1–100.0]	59.0 [49.7–67.8]	2.44 [2.0–3.0]	//	36.7 [26.1–48.3]	100.0 [95.0–100.0]
*PCT*	0.544 [0.452–0.634]	0.484	7.04	60.7 [40.6–78.5]	53.1 [42.7–63.4]	1.30 [0.9–1.9]	0.74 [0.4–1.2]	27.4 [16.9–40.2]	82.3 [70.4–90.9]
*CRP*	0.702 [0.617–0.777]	<0.001	135	74.1 [53.7–88.9]	58.7 [48.9–68.1]	1.79 [1.3–2.5]	0.44 [0.2–0.9]	30.8 [19.9–43.4]	90.1 [80.7–95.9]
*IL-4*	0.689 [0.583–0.782]	0.007	3.34	45.0 [23.1–68.5]	87.1 [77.0–93.9]	3.50 [1.6–7.6]	0.63 [0.4–0.9]	50.9 [26.0–74.0]	74.7 [74.3–92.1]
*IL-6*	0.769 [0.669–0.852]	<0.001	142.13	75.0 [50.9–91.3]	68.6 [56.4–79.1]	2.39 [1.6–3.7]	0.36 [0.2–0.8]	40.5 [24.8–57.9]	90.6 [79.3–96.9]
*IL-8*	0.713 [0.608–0.803]	<0.001	413.48	75.0 [50.9–91.3]	64.3 [51.9–75.4]	2.10 [1.4–3.1]	0.39 [0.2–0.8]	37.5 [22.7–54.2]	90.0 [78.2–96.7]
*IL-10*	0.759 [0.658–0.843]	<0.001	20.17	65.0 [40.8–84.6]	78.6 [67.1–87.5]	3.03 [1.7–5.3]	0.45 [0.2–0.8]	46.4 [27.5–66.1]	88.7 [78.1–95.3]
*TNF-a*	0.663 [0.556–0.759]	0.030	19.83	55.0 [31.5–76.9]	81.4 [70.3–89.7]	2.96 [1.6–5.6]	0.55 [0.3–0.9]	45.8 [25.6–67.2]	86.4 [75.7–93.6]
*IL-35*	0.677 [0.590–0.755]	<0.001	53.97	96.7 [82.8–99.9]	44.7 [34.9–54.8]	1.75 [1.5–2.1]	0.08 [0.0–0.5]	33.7 [23.9–44.7]	97.9 [88.7–99.9]

Data are presented with 95% confidence intervals. All ROC analyses were performed on the entire study population. In the diagnostic ROC analysis, 121 patients were classified as having a positive outcome (bloodstream infection) and 30 as having a negative outcome (no bloodstream infection). In the prognostic ROC analysis, 29 patients had a positive outcome (death) and 122 a negative outcome (survivors). CRP, C-reactive protein; PCT, procalcitonin; IL-4, interleukin-4; IL-6, interleukin-6; IL-8, interleukin-8; IL-10, interleukin-10; IL-35, interleukin-35; TNF-α, tumor necrosis factor-α; PLR, positive likelihood ratio; NLR, negative likelihood ratio; PPV, positive predictive value; NPV, negative predictive value.

## Data Availability

The dataset used and/or analysed during the current study are available from the corresponding author on reasonable request.

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
