# Peer review of "Calprotectin, Azurocidin, and Interleukin-8: Neutrophil Signatures with Diagnostic and Prognostic Value in Sepsis"

_biomedicines, 2025, doi:10.3390/biomedicines13112673_

Round 1
Reviewer 1 Report
Comments and Suggestions for Authors
This multicentre retrospective observational study measured serum Calprotectin, azurocidin, IL-8 and a panel of cytokines (Th1/Th2/Treg/Breg), plus PCT and CRP, in 136 adults (controls, SIRS, culture-positive sepsis, and sepsis deaths) to evaluate diagnostic and prognostic performance. Authors report that azurocidin, Calprotectin, IL-4, IL-6 and IL-8 discriminate sepsis from SIRS and correlate with 28-day mortality, with IL-35 and PCT showing strong diagnostic but limited prognostic value; logistic regression identifies IL-8, azurocidin and Calprotectin as independent predictors of poor outcome. The manuscript combines ANOVA, ROC and logistic models and concludes neutrophil-derived markers add clinically useful information beyond CRP/PCT.
Major revisions
- Methods:
- Provide clear, reproducible inclusion criteria for "suspected systemic organ dysfunction/sepsis" and explicit timing of sample collection relative to symptom onset, hospital admission and blood culture positivity. State whether samples were collected before antibiotic administration.
- Clarify how the SIRS group was defined (which clinical criteria used) and confirm that SIRS patients had negative cultures during the same sampling window.
- Confounding and case-mix: Report comorbidities, sources of infection, antimicrobial therapy, ICU vs ward location, and organ-support requirements for each group. Provide SOFA or at least components (or explain omission) because disease severity is a major confounder for biomarker levels and mortality.
- Statistical analysis details and multiple comparisons
- State how normality was assessed and whether data transformations were applied. Report exact p-values, test statistics and degrees of freedom for ANOVA results.
- Describe adjustment for multiple comparisons (many biomarkers tested) or provide corrected p-values (e.g., Bonferroni or FDR). Without correction, risk of type I error is high.
- For logistic regression models give full model details: covariates included (age, sex, comorbidities, severity scores), continuous vs dichotomous handling of biomarkers, rationale for cutoffs, and model calibration/performance metrics (AUC, Hosmer-Lemeshow or calibration plot).
- ROC analysis transparency: Provide confidence intervals for AUROCs and report how cutoffs were chosen (Youden index, predefined). Include sample sizes used for each ROC and show tabulated sensitivity/specificity values at clinically relevant thresholds.
- Data presentation and completeness: Include a table with baseline characteristics and key laboratory values by group (age, sex, comorbidities, infection source, qSOFA/SOFA, timing of sample, antibiotic exposure). Add the number of missing samples per biomarker and how missing data were handled.
- Interpretation and causality: Tone down causal language. Emphasize that associations do not prove clinical utility until validated prospectively and compared head-to-head with combined clinical scores. Discuss how comorbidity and severity could explain biomarker elevation.
- External validity and sample composition: Report the microorganisms isolated and whether findings apply to gram-positive, gram-negative or fungal sepsis. If the septic cohort is primarily gram-negative, state this as a limitation to generalizability.
Minor revisions
- Title and abstract: ensure consistent terms (e.g., "sepsis" vs "bacteremic sepsis") and add sample sizes per group in the abstract.
- Units and ranges: standardize units across text, tables and figures (e.g., pg/ml, ng/ml, µg/ml) and list assay detection limits and intra/inter-assay CVs for each biomarker.
- Figures and tables: improve readability, e.g. enlarge axis labels, include sample sizes on plots, and provide AUROC values and 95% CIs directly on ROC panels.
- Table 1–3 clarity: ensure all cells contain single-line entries; state number of observations used for each analysis and correct typographical errors (e.g., misplaced numbers or symbols).
- Methods assays: specify vendors, catalogue numbers and whether lab personnel were blinded to clinical status.
- Ethics and data sharing: confirm trial registration if any and provide a statement on availability of anonymized dataset or analytic code.
- Language and style: correct minor English grammar and punctuation issues; streamline the Discussion to avoid repetition.
- Limitations: explicitly state retrospective design, single-timepoint sampling and limited sample size for some subgroup analyses.
Author Response
REVIEWER #1
This multicentre retrospective observational study measured serum Calprotectin, azurocidin, IL-8 and a panel of cytokines (Th1/Th2/Treg/Breg), plus PCT and CRP, in 136 adults (controls, SIRS, culture-positive sepsis, and sepsis deaths) to evaluate diagnostic and prognostic performance. Authors report that azurocidin, Calprotectin, IL-4, IL-6 and IL-8 discriminate sepsis from SIRS and correlate with 28-day mortality, with IL-35 and PCT showing strong diagnostic but limited prognostic value; logistic regression identifies IL-8, azurocidin and Calprotectin as independent predictors of poor outcome. The manuscript combines ANOVA, ROC and logistic models and concludes neutrophil-derived markers add clinically useful information beyond CRP/PCT.
We sincerely thank the Reviewer for the thoughtful and constructive comments. We have carefully addressed all the points raised, including a thorough language revision, with the aim of improving the overall clarity and quality of the manuscript. We hope that the revised version meets the Reviewer’s expectations and that our efforts to enhance both the content and presentation are appreciated.
MAJOR REVISIONS
- Provide clear, reproducible inclusion criteria for "suspected systemic organ dysfunction/sepsis" and explicit timing of sample collection relative to symptom onset, hospital admission and blood culture positivity. State whether samples were collected before antibiotic administration.
We agree with the Reviewer that the manuscript lacked clarity. We have now included the requested information, which is essential for reproducibility.
- Clarify how the SIRS group was defined (which clinical criteria used) and confirm that SIRS patients had negative cultures during the same sampling window.
We sincerely thank the Reviewer for the valuable suggestion. Following her/his recommendation, we have now added the appropriate definitions to clarify these points in the revised manuscript.
- Confounding and case-mix: Report comorbidities, sources of infection, antimicrobial therapy, ICU vs ward location, and organ-support requirements for each group. Provide SOFA or at least components (or explain omission) because disease severity is a major confounder for biomarker levels and mortality.
Done, we have added the required table of patients characteristics (see Table 1).
- State how normality was assessed and whether data transformations were applied. Report exact p-values, test statistics and degrees of freedom for ANOVA results.
We sincerely apppreciate the Reviewer comment. As suggested by the Reviewer, we have now assessed the normality of the data using the Kolmogorov-Smirnov test. Given the non-parametric distribution, data are expressed as median [interquartile range], and the Kruskal–Wallis test was applied (i.e., an ANOVA test on ranks for non-repeated measurements). Post hoc multiple comparisons were performed using Dunn’s test, which includes a built-in correction to control for type I error. Accordingly, exact P values are now reported after correction. Statistical methods, as well as the corresponding results and figures, have been extensively revised according to the new statistical design.
- Describe adjustment for multiple comparisons (many biomarkers tested) or provide corrected p-values (e.g., Bonferroni or FDR). Without correction, risk of type I error is high.
Please refer to the answer above, thanks.
- For logistic regression models give full model details: covariates included (age, sex, comorbidities, severity scores), continuous vs dichotomous handling of biomarkers, rationale for cutoffs, and model calibration/performance metrics (AUC, Hosmer-Lemeshow or calibration plot).
We thank the Reviewer for this valuable comment. Following extensive revision of the manuscript, we have decided to remove the logistic regression analysis for several reasons. First, we believe that this analysis would not provide additional clinically meaningful information beyond what is already presented. Second, the statistical model was based on a partial dataset, as some outliers had to be excluded and only consistent cases were considered, which might have introduced bias. Finally, the limited number of events in our study population restricted the inclusion of multiple covariates, thereby compromising the robustness of the model. Indeed, given the limited number of events, the inclusion of multiple covariates would violate the “10 events-per-variable rule of thumb” for logistic regression. For these reasons, we considered it more appropriate to omit this analysis from the revised version of the manuscript.
- ROC analysis transparency: Provide confidence intervals for AUROCs and report how cutoffs were chosen (Youden index, predefined). Include sample sizes used for each ROC and show tabulated sensitivity/specificity values at clinically relevant thresholds.
Following the Reviewer’s suggestion, we revised the ROC analysis and reported the required data. Accordingly, we modified both the table and the figures, also in the attempt to improve their clarity (see also the answer below).
- Data presentation and completeness: Include a table with baseline characteristics and key laboratory values by group (age, sex, comorbidities, infection source, qSOFA/SOFA, timing of sample, antibiotic exposure). Add the number of missing samples per biomarker and how missing data were handled.
We thank the Reviewer for highlighting this omission. We sincerely apologize for having forgotten to upload and include this table, which had already been prepared. Please note that there were no missing samples for any biomarker, ensuring the consistency of our data.
- Interpretation and causality: Tone down causal language. Emphasize that associations do not prove clinical utility until validated prospectively and compared head-to-head with combined clinical scores. Discuss how comorbidity and severity could explain biomarker elevation.
We thank the Reviewer for this thoughtful point. All the concerns have been carefully addressed. Furthermore, we have added a brief paragraph at the end of the Discussion discussing how comorbidities may contribute to biomarker elevation.
- External validity and sample composition: Report the microorganisms isolated and whether findings apply to gram-positive, gram-negative or fungal sepsis. If the septic cohort is primarily gram-negative, state this as a limitation to generalizability.
As specified in the Materials and Methods section, we included only patients with Gram-negative isolations. As requested by the Reviewer, we have now added this limitation to our study.
Minor revisions
- Title and abstract: ensure consistent terms (e.g., "sepsis" vs "bacteremic sepsis") and add sample sizes per group in the abstract.
Following the Reviewer’s suggestion, we have slightly modified the title to make it more polished. The abstract has also been revised to reflect the updated results.
- Units and ranges: standardize units across text, tables and figures (e.g., pg/ml, ng/ml, µg/ml) and list assay detection limits and intra/inter-assay CVs for each biomarker.
Done, thanks.
- Figures and tables: improve readability, e.g. enlarge axis labels, include sample sizes on plots, and provide AUROC values and 95% CIs directly on ROC panels.
We agree with the Reviewer that the figures required re-editing to improve their clarity. We have revised them accordingly and hope that they are now satisfactory (see also the point above).
- Table 1–3 clarity: ensure all cells contain single-line entries; state number of observations used for each analysis and correct typographical errors (e.g., misplaced numbers or symbols).
All tables have been thoroughly revised to address the Reviewer’s comments. We trust that the revised versions now meet the Reviewer’s requirements.
- Methods assays: specify vendors, catalogue numbers and whether lab personnel were blinded to clinical status.
Done, thanks.
- Ethics and data sharing: confirm trial registration if any and provide a statement on availability of anonymized dataset or analytic code.
We thank the Reviewer for this point. According to the International Committee of Medical Journal Editors (ICMJE), prospective registration is mandatory for all prospective clinical trials involving human participants, in order to promote transparency and prevent unethical conduct. Although a lack of transparency, publication bias, and selective reporting remain major concerns in medical research (see PMID: 21383991), prospective registration helps mitigate these issues - particularly selective reporting - by making key protocol information publicly available before the first participant is enrolled. In contrast, in retrospective study designs, all data are collected after participants have already received treatment, making true prospective trial registration not applicable.
A statement regarding the availability of the anonymized dataset has now been included in the revised manuscript (see first paragraph in Materials and Methods).
- Language and style: correct minor English grammar and punctuation issues; streamline the Discussion to avoid repetition.
The manuscript has been extensively revised in both content and language. See also above.
- Limitations: explicitly state retrospective design, single-timepoint sampling and limited sample size for some subgroup analyses.
Following the Reviewer’s suggestions, we have extensively revised this section of the manuscript and added appropriate references to strengthen and support our discussion.
Reviewer 2 Report
Comments and Suggestions for Authors
The manuscript addresses the diagnostic and prognostic value of neutrophil-associated biomarkers (Calprotectin, Azurocidin, and IL-8), together with several cytokines (IL-4, IL-6, IL-10, IL-35, TNF-α), in patients with sepsis. The topic is clinically relevant and falls well within the scope of Biomedicines. The work is well structured, and the data presentation is generally clear.
I only have minor comments:
- Abstract:
- Clarify the design (“multicenter retrospective observational study”) and state sample sizes for each group (controls, SIRS, septic alive, septic dead).
- Include the main finding succinctly (e.g., “Calprotectin, Azurocidin, and IL-8 showed the strongest combined diagnostic and prognostic performance”).
- Language and Style:
- The manuscript is readable but would benefit from light editing for grammar and phrasing (e.g., “fascinating behaviour” → “notable trend”; “exploit” → “utilize”).
- Avoid colloquial expressions such as “extremely interesting data were found.”
- References:
- The reference list is comprehensive. Ensure consistent formatting according to MDPI style (uniform DOI presentation).
- Figures:
- Provide high-resolution images suitable for print.
Author Response
REVIEWER 2
The manuscript addresses the diagnostic and prognostic value of neutrophil-associated biomarkers (Calprotectin, Azurocidin, and IL-8), together with several cytokines (IL-4, IL-6, IL-10, IL-35, TNF-α), in patients with sepsis. The topic is clinically relevant and falls well within the scope of Biomedicines. The work is well structured, and the data presentation is generally clear.
We sincerely thank the Reviewer for her/his positive and thoughtful feedback. All comments have been carefully considered and incorporated into the revised manuscript, together with the suggestions provided by the other Reviewer.
I only have minor comments:
- Abstract: Clarify the design (“multicenter retrospective observational study”) and state sample sizes for each group (controls, SIRS, septic alive, septic dead).
We have added this data in the abstract. Thanks for your suggestion.
- Abtract: Include the main finding succinctly (e.g., “Calprotectin, Azurocidin, and IL-8 showed the strongest combined diagnostic and prognostic performance”).
Results in the abstract section have been extensively revised, following the suggestions of Reviewer #1.
- Language and Style: the manuscript is readable but would benefit from light editing for grammar and phrasing (e.g., “fascinating behaviour” → “notable trend”; “exploit” → “utilize”).
The manuscript has been extensively revised in contents and form, following also comments by Reviewer 1.
- Avoid colloquial expressions such as “extremely interesting data were found.”
See the previous comment.
- References: The reference list is comprehensive. Ensure consistent formatting according to MDPI style (uniform DOI presentation).
Thanks, references have been nowupdated per Journal Style
- Figures: Provide high-resolution images suitable for print.
Figures have been completely modified, following the new statistical designed reqiuired by Reviewer #1.
Round 2
Reviewer 1 Report
Comments and Suggestions for Authors
The authors’ replies address most of Reviewer 1’s formal points and the manuscript is clearer for methods, tables and figures. Several substantive and transparency issues remain that, if fixed, would strengthen credibility and generalizability.
Major issues to revise or clarify
-
Timing of sampling: The authors state samples were taken “at admission, before antibiotic treatment,” but they must report quantitative timing: median and IQR of interval from symptom onset to sampling, from hospital admission to sampling, and from sampling to blood-culture positivity. These timings affect biomarker interpretation and reproducibility.
-
Case-mix and confounding: Removing the logistic regression is understandable given limited events, but the lack of any multivariable analysis weakens claims of independent predictive value. Recommend at least one of:
-
-
Stratified analyses (e.g., by SOFA tertiles or ICU vs ward) to check consistency of associations.
-
Parsimonious multivariable models (max 2–3 covariates such as age, SOFA and renal dysfunction) with explicit selection criteria.
-
Sensitivity analyses excluding patients with major comorbidities (e.g., chronic kidney disease).
-
-
-
Multiple comparisons and Type I error: Dunn’s post-hoc test addresses pairwise comparisons, but there is still multiplicity across many biomarkers and ROC tests. Authors should:
-
-
Explicitly declare a strategy for global error control or FDR (e.g., Bonferroni or Benjamini–Hochberg) for the prespecified primary hypotheses, and define which biomarkers were primary versus exploratory.
-
If no global correction is applied, justify why findings remain credible (effect sizes, biological consistency).
-
-
-
Statistical transparency: For Kruskal–Wallis report the test statistic (H) and exact p-values in key results. For each post-hoc comparison, ensure corrected p-values are presented in the text for major findings. For ROC analyses, report the N used for each ROC (N positives / N negatives) in the figure captions or panels.
-
Causal language and interpretation: Further tone down causal wording. Keep statements to associations and highlight that clinical utility requires prospective validation and comparison with combined clinical scores.
-
Generalizability: Emphasize early and clearly that the sepsis cohort includes only Gram-negative bloodstream infections and explicitly state this as the main limitation in abstract and discussion.
Author Response
Reviewer #1
The authors’ replies address most of Reviewer 1’s formal points and the manuscript is clearer for methods, tables and figures. Several substantive and transparency issues remain that, if fixed, would strengthen credibility and generalizability.
Major issues to revise or clarify
- Timing of sampling: The authors state samples were taken “at admission, before antibiotic treatment,” but they must report quantitative timing: median and IQR of interval from symptom onset to sampling, from hospital admission to sampling, and from sampling to blood-culture positivity. These timings affect biomarker interpretation and reproducibility.
We thank the Reviewer for this valuable comment. As this was a retrospective study, the exact timing of sampling in relation to symptom onset, hospital admission, and blood culture positivity was not systematically recorded for all patients and therefore cannot be reliably reported as quantitative measures (e.g., median and IQR). However, all samples included in the study were collected at hospital admission, before the initiation of antibiotic treatment, as documented in the patients’ clinical records. This standardized sampling approach minimizes variability related to timing and ensures consistency across the cohort. We acknowledge that precise temporal data would provide additional granularity, and we have now explicitly stated this as a limitation of the study in the revised manuscript. Future prospective studies with predefined time-point documentation will be needed to further explore the influence of sampling timing on biomarker interpretation and reproducibility.
- Case-mix and confounding: Removing the logistic regression is understandable given limited events, but the lack of any multivariable analysis weakens claims of independent predictive value. Recommend at least one of:
- Stratified analyses (e.g., by SOFA tertiles or ICU vs ward) to check consistency of associations.
- Parsimonious multivariable models (max 2–3 covariates such as age, SOFA and renal dysfunction) with explicit selection criteria.
- Sensitivity analyses excluding patients with major comorbidities (e.g., chronic kidney disease).
- Multiple comparisons and Type I error: Dunn’s post-hoc test addresses pairwise comparisons, but there is still multiplicity across many biomarkers and ROC tests.
Authors should explicitly declare a strategy for global error control or FDR (e.g., Bonferroni or Benjamini–Hochberg) for the prespecified primary hypotheses, and define which biomarkers were primary versus exploratory. If no global correction is applied, justify why findings remain credible (effect sizes, biological consistency).
We thank the Reviewer for this comment. In our study, all comparisons were performed using nonparametric tests due to the non-normal distribution of the data. Differences between groups were assessed using the Kruskal–Wallis test, followed by Dunn’s post hoc test with correction for multiple pairwise comparisons, which provides appropriate control of Type I error at the group-comparison level.
Regarding global error control, our analysis focused primarily on a limited set of prespecified biomarkers that have well-established biological relevance in sepsis and infection. These biomarkers were analyzed as primary variables for diagnostic and prognostic evaluation, while no extensive exploratory screening across large panels was performed. Therefore, the risk of inflated false discovery rate (FDR) is limited.
Moreover, the consistency of the results across multiple statistical measures (AUC, likelihood ratios, predictive values) and their biological plausibility within the pathophysiology of sepsis further support the robustness and credibility of the findings, even without an additional global correction such as Bonferroni or Benjamini–Hochberg adjustment.
- Statistical transparency:
- For Kruskal–Wallis report the test statistic (H) and exact p-values in key results.
To preserve the clarity of the manuscript, we have now included the Kruskal–Wallis test statistics (H) in Table 2. The exact p-values were already reported in the table. In addition, as mentioned below, we preferred to avoid excessive redundancy among the text, figures, and tables in order to maintain clarity and readability.
- For each post-hoc comparison, ensure corrected p-values are presented in the text for major findings.
Please note that all exact p-values were reported in the tables and figures, and we prefer to avoid redundancy in the text to ensure clarity and conciseness (as it is required by the editorial indications of the most scientific journals).
- For ROC analyses, report the N used for each ROC (N positives / N negatives) in the figure captions or panels.
Please note also that the numbers of positive and negative samples (N used for each ROC) were already reported in the table captions below each table, where the corresponding ROC data are summarized!
- Causal language and interpretation: Further tone down causal wording. Keep statements to associations and highlight that clinical utility requires prospective validation and comparison with combined clinical scores.
We were quite surprised by this comment, as we do not find any causal wording throughout the manuscript. Moreover, we have already stated in the final sentence that “future prospective, longitudinal studies, for further validation, with larger and more balanced cohorts to confirm and expand upon our results”. Therefore, we believe this point has been adequately addressed, and we are unsure what additional clarification the Reviewer is requesting.
- Generalizability: Emphasize early and clearly that the sepsis cohort includes only Gram-negative bloodstream infections and explicitly state this as the main limitation in abstract and discussion.
The point raised by the Reviewer refers more to reproducibility rather than generalizability. The specific inclusion and exclusion criteria were already clearly described in the manuscript, addressing this aspect comprehensively. In particular, the definition of the sepsis patient cohort, including the inclusion criteria for Gram-negative bloodstream infection, was provided in the Materials and Methods section after Revision 1! It is also worth noting that generalizability is typically discussed in the Limitations section, as we have already did in Revision 1.
As authors we are also quite surprised by the Reviewer’s request to insert limitations in the abstract. Most of the authors serve on editorial boards of several Q1-ranked journals and have published in leading journals such as The New England Journal of Medicine, JAMA, and The Lancet. During our experience as authors, we never had before such a request from a Reviewer and never asked this as Editors. The aim of an abstract is far from discussing limitations of a study, rather to provide a quick and clear idea of the manuscript contents/findings. Therefore, we do not accept this concern by the Reviewer.